# LEARNING TO PROMPT FOR VISION-LANGUAGE MODELS

## ABSTRACT

Vision-language pre-training has recently emerged as a promising alternative for representation learning. It shifts from the tradition of using images and discrete labels for learning a fixed set of weights, seen as visual concepts, to aligning images and raw text for two separate encoders. Such a paradigm benefits from a broader source of supervision and allows zero-shot transfer to downstream tasks since visual concepts can be diametrically generated from natural language, known as prompt. In this paper, we identify that a major challenge of deploying such models in practice is prompt engineering. This is because designing a proper prompt, especially for context words surrounding a class name, requires domain expertise and typically takes a significant amount of time for words tuning since a slight change in wording could have a huge impact on performance. Moreover, different downstream tasks require specific designs, further hampering the efficiency of deployment. To overcome this challenge, we propose a simple approach named *context optimization (CoOp)*. The main idea is to model context in prompts using continuous representations and perform end-to-end learning from data while keeping the pre-trained parameters fixed. In this way, the design of task-relevant prompts can be fully automated. Experiments on 11 datasets show that CoOp effectively turns pre-trained vision-language models into data-efficient visual learners, requiring as few as one or two shots to beat hand-crafted prompts with a decent margin and able to gain significant improvements when using more shots (e.g., at 16 shots the average gain is around 17% with the highest reaching over 50%). CoOp also exhibits strong robustness to distribution shift.

## 1 INTRODUCTION

The traditional approach for visual representation learning is to train vision models to predict for a fixed set of object categories using discrete labels (He et al., 2016; Dosovitskiy et al., 2021). However, this approach limits visual recognition systems to closed-set visual concepts defined during training, making them unable to handle new categories once deployed in target environments, since additional data are required for learning a new classifier. Recently, vision-language pre-training such as CLIP (Radford et al., 2021) and ALIGN (Jia et al., 2021) has emerged as a promising alternative. The main idea is to align images and raw text using two separate encoders—one for each modality. Through large-scale pre-training, vision-language models are allowed to learn open-set visual concepts and can readily be transferred to downstream tasks. In particular, for each new classification task, one can synthesize the classification weights by feeding natural language describing classes of interest to the text encoder, and compare them with image features produced by the image encoder.

We observe that for pre-trained vision-language models, the text input, known as prompt, plays a key role in downstream datasets. However, identifying the right prompt is a non-trivial task, which often takes a significant amount of time for words tuning—since a slight change in wording could make a huge difference in performance. For instance, for Caltech101 (Figure 1(a), 2nd vs. 3rd prompt), adding "a" before the class token brings more than 5% increase in accuracy. Moreover, prompt engineering also requires expertise about the task and ideally the language model's underlying mechanism. This is exemplified in Figure 1(b-d) where adding task-relevant context can lead to significant improvements, i.e., "flower" for Flowers102, "texture" for DTD and "satellite" for EuroSAT. Tuning the sentence structure could bring further improvements, e.g., putting "a type of flower" after the class token for Flowers102, keeping only "texture" in the context for DTD, and

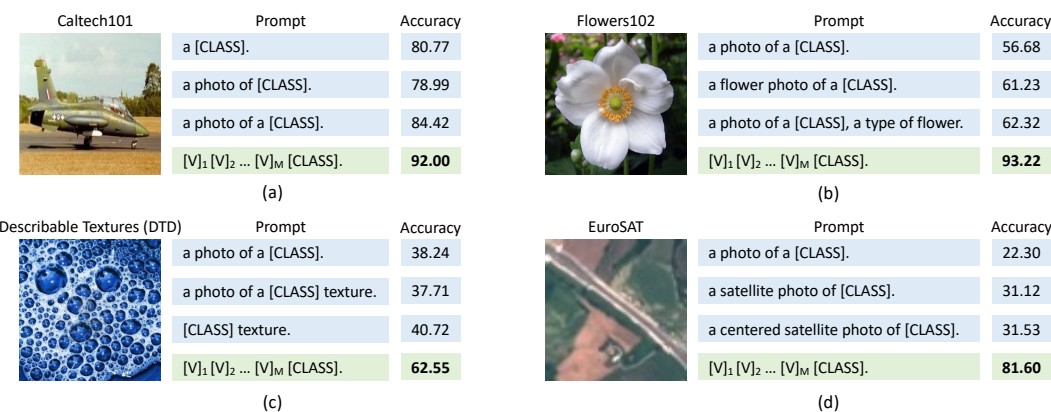

Figure 1: **Prompt engineering vs. context optimization (CoOp)**. The latter uses only 16 shots for learning in these examples.

adding "centered" before "satellite photo" for EuroSAT. However, even with extensive tuning, the resulting prompts are by no means guaranteed to be optimal for these downstream tasks.

Inspired by recent prompt learning research in NLP (Shin et al., 2020; Jiang et al., 2020; Zhong et al., 2021), we propose *context optimization (CoOp)*[1] to automate prompt engineering to allow more efficient and task-specific transfer for pre-trained vision-language models. Specifically, we model a prompt's context using continuous representations which are essentially initialized with random vectors with the same dimension as word embeddings (see Figure 2). The context could be shared among all classes or designed to be class-specific. During training, we simply minimize the prediction error using the cross-entropy loss with respect to the learnable context vectors, while keeping the pre-trained parameters fixed. The gradients can be back-propagated all the way through the text encoder, distilling the rich knowledge encoded in the parameters for learning task-relevant context.

To demonstrate the effectiveness of CoOp, we benchmark on 11 datasets, which cover a diverse set of visual recognition tasks including classification on generic objects, scenes, actions and fine-grained categories, as well as specialized tasks like recognizing textures and satellite imagery. The results show that CoOp can effectively turn pre-trained vision-language models into data-efficient visual learners, requiring as few as one or two shots to beat hand-crafted prompts with a decent margin. The performance can also be further boosted by using more shots, e.g., at 16 shots the margin over hand-crafted prompts averages at around 17% and reaches over 50% for the highest. CoOp also outperforms the linear probe alternative known as a strong few-shot learning baseline (Tian et al., 2020), and crucially, demonstrates much stronger robustness to distribution shift. Extensive analysis is also conducted to offer a comprehensive picture on how to apply CoOp in practice. The source code for reproducing the experiments will be released to facilitate future research.

## 2 METHODOLOGY

### 2.1 VISION-LANGUAGE PRE-TRAINING

We briefly introduce vision-language pre-training with a particular focus on CLIP (Radford et al., 2021). Our approach is applicable to broader CLIP-like vision-language models.

**Models** CLIP consists of two encoders, one for images and the other for text. The image encoder aims to map high-dimensional images into a low-dimensional embedding space. The architecture of the image encoder can take the form of a CNN like ResNet-50 (He et al., 2016) or a ViT (Dosovitskiy et al., 2021). On the other hand, the text encoder is built on top of a Transformer (Vaswani et al., 2017) and aims to generate text representations from natural language.

---

[1]CoOp is pronounced as /ku:p/.

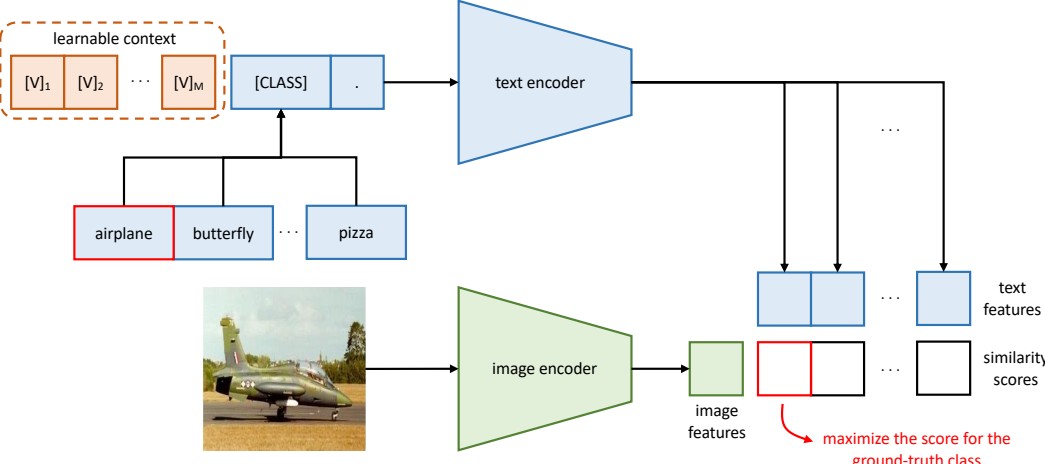

Figure 2: Overview of context optimization (CoOp).

Specifically, given a sequence of words (tokens), such as "a photo of a dog.", CLIP first converts each one of the token (including punctuation) into a lower-cased byte pair encoding (BPE) representation (Sennrich et al., 2016), which is essentially a unique numeric ID. The vocabulary size in CLIP is 49,152. To facilitate minibatch processing, each text sequence is encompassed with the [SOS] and [EOS] tokens and capped at a fixed length of 77. After that, the IDs are mapped to 512-D word embedding vectors, which are then passed on to the Transformer. Finally, the features at the [EOS] token position are layer normalized and further processed by a linear projection layer.

**Training**  CLIP is trained to align the two embedding spaces learned for images and text respectively. Specifically, the learning objective is formulated as a contrastive loss. Given a batch of image-text pairs, CLIP maximizes the cosine similarity for matched pairs while minimizes the cosine similarity for all other unmatched pairs. To learn diverse visual concepts that are more transferable to downstream tasks, CLIP's team collects a large training dataset consisting of 400 million image-text pairs.

**Zero-Shot Inference**  Since CLIP is pre-trained to predict whether an image matches a textual description, it naturally fits zero-shot recognition. This is achieved by comparing image features with the classification weights synthesized by the text encoder, which takes as input textual descriptions specifying classes of interest. Formally, let $\boldsymbol{f}$ be image features extracted by the image encoder for an image $\boldsymbol{x}$ and $\{\boldsymbol{w}_i\}_{i=1}^{K}$ a set of weight vectors generated by the text encoder. $K$ denotes the number of classes and each $\boldsymbol{w}_i$ is derived from a prompt that could have the form of "a photo of a [CLASS]." where the class token is replaced by the specific class name, such as "cat", "dog" or "car". The prediction probability is then computed as

$$p(y = i|\boldsymbol{x}) = \frac{\exp(<\boldsymbol{w_i}, \boldsymbol{f}>/\tau)}{\sum_{j=1}^{K}\exp(<\boldsymbol{w_j}, \boldsymbol{f}>/\tau)}, \tag{1}$$

where $\tau$ is a temperature parameter learned by CLIP and $<\cdot, \cdot>$ denotes cosine similarity.

## 2.2 CONTEXT OPTIMIZATION

We propose context optimization (CoOp), which avoids manual prompt tuning by modeling context words with continuous vectors that are end-to-end learned from data. An overview is shown in Figure 2. Specifically, the prompt given to the text encoder $g(\cdot)$ is designed with the following form,

$$\boldsymbol{t} = [\text{V}]_1[\text{V}]_2 \ldots [\text{V}]_M[\text{CLASS}], \tag{2}$$

where each $[\text{V}]_m$ ($m \in \{1, \ldots, M\}$) is a vector with the same dimension as word embeddings (i.e., 512 for CLIP), and $M$ is a hyperparameter specifying the number of context tokens. Note that the

context here is *shared* among all classes, which is called unified context and different from class-specific context that is introduced later.

By forwarding a prompt $t$ to the text encoder $g(\cdot)$, we can obtain a classification weight vector representing a visual concept. The prediction probability is computed as

$$p(y = i | \boldsymbol{x}) = \frac{\exp(< g(\boldsymbol{t}_i), \boldsymbol{f} > / \tau)}{\sum_{j=1}^{K} \exp(< g(\boldsymbol{t}_j), \boldsymbol{f} > / \tau)}, \tag{3}$$

where the class token within each prompt $\boldsymbol{t}_i$ is replaced by the corresponding word embedding vector(s) of the $i$-th class name.

Training is performed to minimize the standard classification loss based on the cross-entropy, and the gradients can be back-propagated all the way through the text encoder $g(\cdot)$, making use of the rich knowledge encoded in the parameters to optimize the context. The design of continuous representations also allows full exploration in the word embedding space, which facilitates the learning of task-relevant context.

**Other Variants**   Other than placing the class token at the end of a sequence as in Equation (2), we can also put it in the middle like

$$\boldsymbol{t} = [\text{V}]_1 \ldots [\text{V}]_{\frac{M}{2}} [\text{CLASS}][\text{V}]_{\frac{M}{2}+1} \ldots [\text{V}]_M, \tag{4}$$

which increases flexibility for learning—theoretically, the prompt is allowed to either fill the latter cells with supplementary descriptions or cut off the sentence earlier by using a termination signal such as full stop.

Another option is to design class-specific context (CSC) where context vectors are independent to each class, i.e., $[\text{V}]_1^i [\text{V}]_2^i \ldots [\text{V}]_M^i \neq [\text{V}]_1^j [\text{V}]_2^j \ldots [\text{V}]_M^j$ for $i \neq j$ and $i, j \in \{1, \ldots, K\}$. As an alternative to unified context, we find that CSC is particularly useful for some fine-grained classification tasks.

## 3 EXPERIMENTS

### 3.1 FEW-SHOT LEARNING

**Datasets**   We select 11 publicly available image classification datasets used in CLIP: ImageNet (Deng et al., 2009), Caltech101 (Fei-Fei et al., 2004), OxfordPets (Parkhi et al., 2012), StanfordCars (Krause et al., 2013), Flowers102 (Nilsback & Zisserman, 2008), Food101 (Bossard et al., 2014), FGVCAircraft (Maji et al., 2013), SUN397 (Xiao et al., 2010), DTD (Cimpoi et al., 2014), EuroSAT (Helber et al., 2019) and UCF101 (Soomro et al., 2012) (see Appendix A for their statistics). These datasets constitute a comprehensive benchmark, which covers a diverse set of vision tasks including classification on generic objects, scenes, actions and fine-grained categories, as well as specialized tasks like recognizing textures and satellite imagery. We follow the few-shot evaluation protocol adopted in CLIP (Radford et al., 2021), using 1, 2, 4, 8 and 16 shots for training respectively and deploying models in the full test sets. The average results over three runs are reported for comparison.

**Training Details**   CoOp has four versions: positioning the class token in the end or middle; unified context vs. CSC. Unless otherwise stated, ResNet-50 (He et al., 2016) is used as the image encoder's backbone and the number of context tokens $M$ is set to 16. Investigations on other design choices are discussed in Section 3.3. All models are built on top of the open-sourced CLIP.[2] CoOp's context vectors are randomly initialized by drawing from a zero-mean Gaussian distribution with standard deviation equal to 0.02. Training is done with SGD and an initial learning rate of 0.002, which is decayed by the cosine annealing rule. The maximum epoch is set to 200 for 16/8 shots, 100 for 4/2 shots, and 50 for 1 shot (except for ImageNet where the maximum epoch is fixed to 50). To mitigate explosive gradients observed in the early training iterations, we use the warmup trick by fixing the learning rate to $1e-5$ during the first epoch.

---

[2]https://github.com/openai/CLIP.

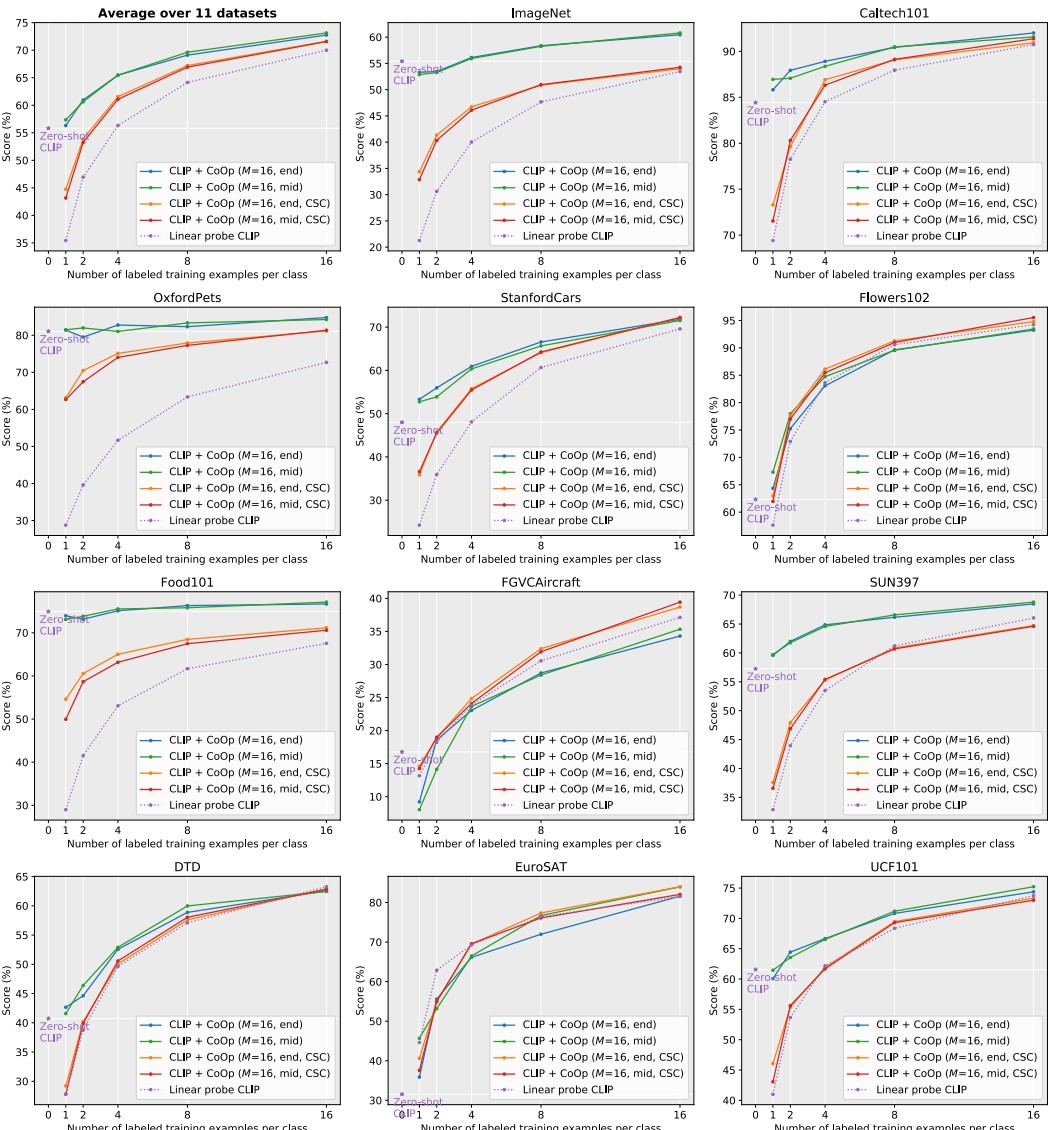

Figure 3: **Main results of few-shot learning on the 11 datasets**. Overall, CoOp effectively turns CLIP into a strong few-shot learner (solid lines), achieving significant improvements over zero-shot CLIP (stars) and performing favorably against the linear probe alternative (dashed lines). $M$ denotes the context length. "end" or "mid" means putting the class token in the end or middle. CSC means class-specific context.

**Baseline Methods**   We compare CoOp with two baseline methods. The first is zero-shot CLIP, which is based on hand-crafted prompts. We follow the guideline of prompt engineering introduced by Radford et al. (2021). For generic objects and scenes, "a photo of a [CLASS]." is adopted. For fine-grained categories, task-relevant context is added like "a type of pet" for OxfordPets and "a type of food" for Food101. When it comes to specialized tasks such as recognizing textures in DTD, the prompt is customized as "[CLASS] texture." where the class names are adjectives like "bubbly" and "dotted". See Appendix A for the details. The second baseline is linear probe CLIP. As suggested by Radford et al. (2021) and a recent study on few-shot learning (Tian et al., 2020), training a linear classifier on top of high-quality pre-trained models' features (like CLIP) can easily achieve performance that is on a par with that of state-of-the-art few-shot learning methods, which are often much more sophisticated. We follow the same training method used by Radford et al. (2021) to train linear probe CLIP.

**Comparison with Hand-Crafted Prompts** Figure 3 summarizes the results. Our default model is CLIP+CoOp with the class token positioned in the end. The two different ways of positioning the class token achieve similar performance as their curves highly overlap. From the average performance displayed in the top-left corner, we observe that CLIP+CoOp is a strong few-shot learner, requiring only two shots on average to obtain a decent margin over zero-shot CLIP. Given 16 shots for training, the average gap brought by CoOp can be further increased to around 17%.

Figure 4 ranks the absolute improvements obtained by CoOp at 16 shots over hand-crafted prompts. Huge improvements are observed on specialized tasks namely EuroSAT and DTD where the increase in performance reaches over 50% and 20% respectively. The jumps in performance are also significant (those more than 10%) on most fine-grained datasets including Flowers102, Stanford-Cars and FGVCAircraft, as well as on scene and action recognition datasets (SUN397 & UCF101). Since ImageNet is a challenging dataset that contains 1,000 classes, the 5.05% improvement is also noteworthy. In contrast, the increases on the two fine-grained datasets, OxfordPets and Food101, are less appealing. By digging into CLIP+CoOp's curves on these two datasets in Figure 3, we find there is a loss of momentum in performance improvements even with more shots used, seemingly an overfitting problem. A potential solution is to impose higher regularizations like increasing the weight decay. Nonetheless, the overall results are strong enough to serve as evidence of CoOp's capability of learning task-relevant prompts in a data-efficient way.

**Comparison with Linear Probe CLIP** In terms of the overall performance (Figure 3, top-left), CLIP+CoOp demonstrates clear advantages over linear probe CLIP. The latter requires 4 shots on average to match the zero-shot's performance while CoOp's average gains at 4 shots are already more than 10%. It is also clear that the gaps in the extreme low-data regime such as one or two shots are much larger, suggesting that CoOp is much more effective than learning a linear classifier from scratch for few-shot learning. We also observe that linear probe CLIP is comparable to CLIP+CoOp on the two specialized tasks (DTD & EuroSAT) as well as on a couple of fine-grained datasets (Flowers102 & FGVCAircraft)—this is not too surprising as the pre-trained CLIP space has been

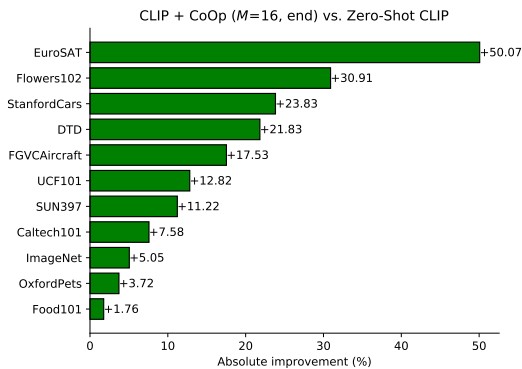

Figure 4: Comparison with hand-crafted prompts.

proved powerful, making the linear probe model a strong competitor. Nevertheless, CoOp's CSC version can beat linear probe CLIP on the aforementioned datasets, and moreover, shows much better potential when more shots become available.

**Unified vs. Class-Specific Context** On average, using unified context leads to better performance. In terms of when to apply CSC and when not to, we have the following suggestions. For generic objects (ImageNet & Caltech101), scenes (SUN397) and actions (UCF101), using unified context is clearly better. Unified context also works better on some fine-grained datasets including OxfordPets and Food101, but on others like StanfordCars, Flowers102 and FGVCAircraft the CSC version is preferred. CSC also yields better performance on the two specialized tasks, DTD and EuroSAT, at 16 shots in particular. However, CSC mostly underperforms unified context in challenging low-data scenarios (fewer than 8 shots), which makes sense because CSC has more parameters than unified context and needs more data for training.

## 3.2 ROBUSTNESS TO DISTRIBUTION SHIFT

Since CoOp requires training on a specific data distribution, it risks learning spurious correlations that are detrimental to generalization in unseen distributions (domains), as suggested in recent studies (Taori et al., 2020; Zhou et al., 2021). On the contrary, zero-shot CLIP is not tied to a specific data distribution and has exhibited strong robustness to distribution shift (Radford et al., 2021). In this section, we aim to unveil how robust CoOp is to distribution shift, in comparison to zero-shot CLIP and the linear probe model.

Table 1: Evaluation on robustness to distribution shift. $M$: CoOp's context length.

| | Source | Target | | | |
|---|---|---|---|---|---|
| | ImageNet | ImageNetV2 | ImageNet-Sketch | ImageNet-A | ImageNet-R |
| Zero-Shot CLIP | 55.41 | 48.08 | 31.67 | 18.63 | 53.45 |
| Linear Probe CLIP | 53.44 | 43.40 | 17.63 | 11.66 | 32.63 |
| CLIP + CoOp ($M=16$) | 60.46 | 52.17 | 31.14 | 19.62 | 53.31 |
| CLIP + CoOp ($M=8$) | **60.90** | 52.53 | 31.73 | 19.97 | 54.34 |
| CLIP + CoOp ($M=4$) | 60.85 | **53.02** | **32.99** | **20.69** | **55.57** |

Table 2: Comparison with prompt ensembling.

| | ImageNet |
|---|---|
| Prompt engineering | 55.41 |
| Prompt ensembling | 57.81 |
| CoOp | **60.46** |

Table 3: Random vs. manual initialization.

| | Avg % |
|---|---|
| $[V]_1[V]_2[V]_3[V]_4$ | 71.26 |
| "a photo of a" | **71.51** |

**Datasets** The source dataset is ImageNet. The target datasets are ImageNetV2 (Recht et al., 2019), ImageNet-Sketch (Wang et al., 2019), ImageNet-A (Hendrycks et al., 2021b) and ImageNet-R (Hendrycks et al., 2021a), all of which have compatible class names with ImageNet allowing seamless transfer for the prompts learned by CoOp. ImageNetV2 is a reproduced test set using different sources while following ImageNet's data collection process. ImageNet-Sketch contains sketch images belonging to the same 1,000 ImageNet classes. Both ImageNet-A and -R contain 200 classes derived from a subset of ImageNet's 1,000 classes. The former consists of real-world adversarially filtered images that cause current ImageNet classifiers to produce low results, whereas the latter features a rendition of the ImageNet classes in diverse image styles such as paintings, cartoons and sculptures.

**Results** Table 1 summarizes the results. It is surprising that CLIP+CoOp exhibits stronger robustness than zero-shot CLIP to distribution shift, despite exposure to the source dataset. This suggests that the learned prompts are also generalizable. Moreover, it is interesting to see that using fewer context tokens leads to better robustness. More results with different vision backbones are provided in Appendix B.1 where the conclusion remains the same. In contrast, linear probe CLIP obtains much worse results on these target datasets, exposing its weakness in domain generalization.

### 3.3 FURTHER ANALYSIS

**Comparison with Prompt Ensembling** Radford et al. (2021) have suggested that additional improvements can be obtained by ensembling over multiple zero-shot classifiers generated using different hand-crafted prompts, such as "a photo of the large [CLASS].", "a bad photo of the [CLASS]." and "a origami [CLASS].", which reflect a different scale, view and abstraction respectively for an image. We are interested to know whether the prompts learned by CoOp can still maintain advantages when compared with prompt ensembling. For fair comparison, we use the select prompts from Radford et al. (2021), which have been extensively tuned on ImageNet, to construct the ensemble classifier. Table 2 presents the results of prompt engineering (i.e., using a single hand-crafted prompt), prompt ensembling and CoOp, confirming that CoOp is still the best performing method. Additional results are provided in Appendix B.2 to show that CoOp also beats prompt ensembling when more advanced vision backbones are used. Given the potential of prompt ensembling, future work could investigate how to improve CoOp from the ensembling perspective.

**Context Length** How many context tokens should be used? And is it better to have more context tokens? The results in Section 3.2 suggest having shorter context length benefits domain generalization. Here we study this hyperparameter for source datasets. Specifically, we repeat experiments on the 11 datasets by varying the context length from 4 to 8 to 16. The average results are shown in Figure 5(a), which indicate that having more context tokens leads to better performance and that positioning the class token in the middle gains more momentum with longer context length. To sum

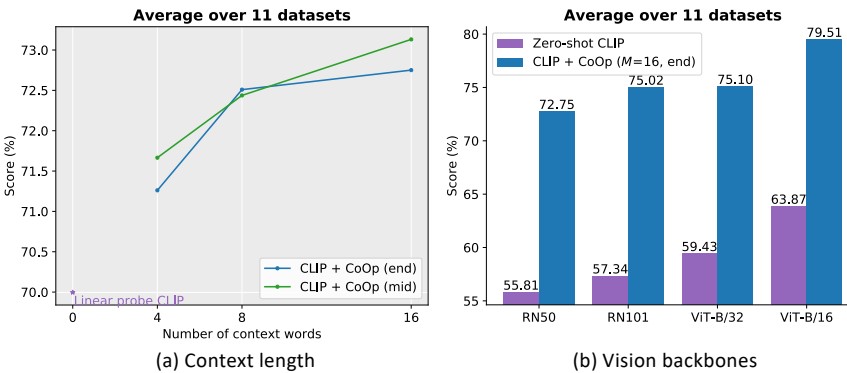

Figure 5: Investigations on CoOp's context length and various vision backbones.

up, there is no golden rule for selecting perfect context length since one needs to balance between performance and robustness to distribution shift. See Appendix B.3 for the dataset-specific results.

**Vision Backbones**  Figure 5(b) summarizes the results on the 11 datasets using a variety of vision backbones covering both CNNs and ViTs. The results are expected: the more advanced the backbone, the better the performance. The gap between CoOp and hand-crafted prompts is significant across all architectures. See Appendix B.4 for the dataset-specific results.

**Initialization**  We compare random initialization with manual initialization. The latter uses the embeddings of "a photo of a" to initialize the context vectors for the 11 datasets. For fair comparison, we also set the context length to 4 when using random initialization. Table 3 suggests a "good" initialization only brings a small improvement. Though further tuning of the initialization words might help, in practice we suggest using the simple random initialization method.

**Interpreting the Learned Prompts**  is difficult because the context vectors are optimized in a continuous space. We resort to an indirect way by searching within the vocabulary for words that are closest to the learned vectors based on the Euclidean distance. Note that CLIP (Radford et al., 2021) uses the BPE representation (Sennrich et al., 2016) for tokenization, so the vocabulary includes subwords that frequently appear in text, such as "hu" (subsumed by many words like "hug" and "human"). Table 4 shows the searched results on some datasets. We observe that a few words are somewhat relevant to the tasks, such as "enjoyed" for Food101, "fluffy" and "paw" for OxfordPets, and "pretty" for DTD. But when connecting all the nearest words together, the prompts do not make much sense. We also observe that when using manual initialization (like "a photo of a"), the nearest words for the converged vectors are mostly the ones used for initialization. We conjecture that the learned vectors might encode meanings that are beyond the existing vocabulary. Overall, we are unable to draw any firm conclusion based on the observations because using nearest words to interpret the learned prompts could be inaccurate—the semantics of the vectors is not necessarily correlated with the nearest words.

## 4  RELATED WORK

**Vision-Language Models**  have recently demonstrated great potential in learning generic visual representations and allowing zero-shot transfer to a variety of downstream classification tasks (Radford et al., 2021; Jia et al., 2021; Zhang et al., 2020). To our knowledge, the recent developments in vision-language learning, particularly CLIP (Radford et al., 2021) and ALIGN (Jia et al., 2021), are largely driven by advances in the following three areas: i) text representation learning with Transformers (Vaswani et al., 2017), ii) large-minibatch contrastive representation learning (Chen et al., 2020; He et al., 2020; Hénaff et al., 2020), and iii) web-scale training datasets—CLIP benefits from 400 million curated image-text pairs while ALIGN exploits 1.8 billion noisy image-text pairs.

The idea of mapping images and text onto a common embedding space has been studied since nearly a decade ago (Socher et al., 2013; Frome et al., 2013; Elhoseiny et al., 2013), but with drastically

Table 4: The nearest words for each of the 16 context vectors learned by CoOp, with their distances shown in parentheses. N/A means non-Latin characters.

| # | ImageNet | Food101 | OxfordPets | DTD | UCF101 |
|---|---|---|---|---|---|
| 1 | potd (1.7136) | lc (0.6752) | tosc (2.5952) | boxed (0.9433) | meteorologist (1.5377) |
| 2 | that (1.4015) | enjoyed (0.5305) | judge (1.2635) | seed (1.0498) | exe (0.9807) |
| 3 | filmed (1.2275) | beh (0.5390) | fluffy (1.6099) | anna (0.8127) | parents (1.0654) |
| 4 | fruit (1.4864) | matches (0.5646) | cart (1.3958) | mountain (0.9509) | masterful (0.9528) |
| 5 | ,... (1.5863) | nytimes (0.6993) | harlan (2.2948) | eldest (0.7111) | fe (1.3574) |
| 6 | ° (1.7502) | prou (0.5905) | paw (1.3055) | pretty (0.8762) | thof (1.2841) |
| 7 | excluded (1.2355) | lower (0.5390) | incase (1.2215) | faces (0.7872) | where (0.9705) |
| 8 | cold (1.4654) | N/A | bie (1.5454) | honey (1.8414) | kristen (1.1921) |
| 9 | stery (1.6085) | minute (0.5672) | snuggle (1.1578) | series (1.6680) | imam (1.1297) |
| 10 | warri (1.3055) | ∼ (0.5529) | along (1.8298) | coca (1.5571) | near (0.8942) |
| 11 | marvelcomics (1.5638) | well (0.5659) | enjoyment (2.3495) | moon (1.2775) | tummy (1.4303) |
| 12 | .: (1.7387) | ends (0.6113) | jt (1.3726) | lh (1.0382) | hel (0.7644) |
| 13 | N/A | mis (0.5826) | improving (1.3198) | won (0.9314) | boop (1.0491) |
| 14 | lation (1.5015) | somethin (0.6041) | srsly (1.6759) | replied (1.1429) | N/A |
| 15 | muh (1.4985) | seminar (0.5274) | asteroid (1.3395) | sent (1.3173) | facial (1.4452) |
| 16 | .# (1.9340) | N/A | N/A | piedmont (1.5198) | during (1.1755) |

different technologies. For text features extraction, early work has mainly utilized pre-trained word vectors (Socher et al., 2013; Frome et al., 2013) or the hand-crafted TF-IDF features (Elhoseiny et al., 2013; Lei Ba et al., 2015). Matching images and text features has been formulated as metric learning (Frome et al., 2013), multi-label classification (Joulin et al., 2016; Gomez et al., 2017), n-gram language learning (Li et al., 2017), and the recently proposed captioning (Desai & Johnson, 2021). Our work is orthogonal to recent research in vision-language models, aiming to facilitate the deployment of such models in downstream datasets.

**Prompt Learning in NLP** Knowledge probing for large pre-trained language models, formally defined by Petroni et al. (2019) as "fill-in-the-blank" cloze tests, has recently sparked interest in prompt learning research in NLP (Shin et al., 2020; Jiang et al., 2020; Li & Liang, 2021; Zhong et al., 2021; Lester et al., 2021; Gao et al., 2020; Liu et al., 2021b). The basic idea of knowledge probing is to induce pre-trained language models to generate answers given cloze-style prompts, which can benefit a number of downstream tasks, such as sentiment analysis. Jiang et al. (2020) propose to generate candidate prompts through text mining and paraphrasing, and identify the optimal ones that give the highest training accuracy. Shin et al. (2020) introduce a gradient-based approach, which searches for tokens with the largest gradient changes in the label likelihood. Most related to our work are continuous prompt learning methods (Zhong et al., 2021; Li & Liang, 2021; Lester et al., 2021) which optimize continuous vectors in the word embedding space. A drawback of such methods compared to searching discrete tokens is the lack of a clear way to visualize what "words" are learned for the vectors. We refer readers to Liu et al. (2021a) for a comprehensive survey in the topic of prompt learning in NLP.

## 5 CONCLUSION

We present CoOp, a differentiable approach that focuses on continuous prompt learning to facilitate the deployment of pre-trained vision-language models in downstream datasets. The results on the 11 datasets serve as strong evidence of CoOp's effectiveness in data-efficient learning. The learned prompts are proved much more task-relevant than hand-crafted prompts reflected by the huge improvements in performance, as well as stronger in robustness to distribution shift. In terms of limitation, CoOp requires explicit access to the pre-trained model parameters, which might be unavailable when only the APIs of pre-trained models are provided. An interesting future direction is thus to investigate "black-box" prompt learning.

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

APPENDIX

# A  DATASETS DETAILS

The detailed statistics of the 11 datasets, as well as the four variants of ImageNet, are shown in Table 5. The hand-crafted prompts used for zero-shot CLIP are also detailed in the table. For Caltech101, the "BACKGROUND_Google" and "Faces_easy" classes are discarded. For the video dataset, UCF101, the middle frame of each video is used as input to the image encoder.

Table 5: Datasets statistics.

| Dataset | Classes | Train | Val | Test | Hand-crafted prompt |
|---------|---------|-------|-----|------|---------------------|
| ImageNet | 1,000 | 1.28M | N/A | 50,000 | "a photo of a [CLASS]." |
| Caltech101 | 100 | 4,128 | 1,649 | 2,465 | "a photo of a [CLASS]." |
| OxfordPets | 37 | 2,944 | 736 | 3,669 | "a photo of a [CLASS], a type of pet." |
| StanfordCars | 196 | 6,509 | 1,635 | 8,041 | "a photo of a [CLASS]." |
| Flowers102 | 102 | 4,093 | 1,633 | 2,463 | "a photo of a [CLASS], a type of flower." |
| Food101 | 101 | 50,500 | 20,200 | 30,300 | "a photo of [CLASS], a type of food." |
| FGVCAircraft | 100 | 3,334 | 3,333 | 3,333 | "a photo of a [CLASS], a type of aircraft." |
| SUN397 | 397 | 15,880 | 3,970 | 19,850 | "a photo of a [CLASS]." |
| DTD | 47 | 2,820 | 1,128 | 1,692 | "[CLASS] texture." |
| EuroSAT | 10 | 13,500 | 5,400 | 8,100 | "a centered satellite photo of [CLASS]." |
| UCF101 | 101 | 7,639 | 1,898 | 3,783 | "a photo of a person doing [CLASS]." |
| ImageNetV2 | 1,000 | N/A | N/A | 10,000 | "a photo of a [CLASS]." |
| ImageNet-Sketch | 1,000 | N/A | N/A | 50,889 | "a photo of a [CLASS]." |
| ImageNet-A | 200 | N/A | N/A | 7,500 | "a photo of a [CLASS]." |
| ImageNet-R | 200 | N/A | N/A | 30,000 | "a photo of a [CLASS]." |

# B  ADDITIONAL RESULTS

## B.1  ROBUSTNESS EXPERIMENTS

In addition to ResNet-50, we further experiment with more advanced architectures including ResNet-101, ViT-B/32 and ViT-B/16, all of which have pre-trained weights available from CLIP's GitHub repository. The results are shown in Table 6 where we can draw the same conclusion as the main paper: CoOp offers stronger robustness than hand-crafted prompts and using fewer context tokens benefits domain generalization.

## B.2  PROMPT ENGINEERING, PROMPT ENSEMBLING AND CoOp

Table 7 provides more comprehensive comparisons covering a variety of vision backbones. The observations are similar to those discussed in the main paper: prompt ensembling is clearly better than prompt engineering; and CoOp demonstrates consistent advantages over prompt ensembling.

## B.3  CONTEXT LENGTH

Figure 6 shows detailed results of using different context lengths for CoOp on each of the 11 datasets. The average performance, displayed in the top-left corner, suggests that using more context tokens is better. There are three exceptions: on ImageNet, OxfordPets, and Food101, the performance is saturated and the improvements diminish when the context length is increased. As discussed in the main paper, selecting a proper length needs to balance between performance on source datasets and robustness to distribution shift in unseen domains. We suggest practitioners use a validation set to identify the optimal context length for their applications.

Table 6: Comparison with zero-shot CLIP on robustness to distribution shift using different vision backbones. $M$: CoOp's context length.

| Method | Source | Target | | | |
|---|---|---|---|---|---|
| | ImageNet | ImageNetV2 | ImageNet-Sketch | ImageNet-A | ImageNet-R |
| **ResNet-50** | | | | | |
| Zero-Shot CLIP | 55.41 | 48.08 | 31.67 | 18.63 | 53.45 |
| CLIP + CoOp ($M=16$) | 60.46 | 52.17 | 31.14 | 19.62 | 53.31 |
| CLIP + CoOp ($M=4$) | **60.85** | **53.02** | **32.99** | **20.69** | **55.57** |
| **ResNet-101** | | | | | |
| Zero-Shot CLIP | 58.72 | 51.57 | 36.73 | 25.11 | 62.15 |
| CLIP + CoOp ($M=16$) | **64.39** | 55.00 | 37.54 | 26.31 | 61.73 |
| CLIP + CoOp ($M=4$) | 63.99 | **55.45** | **39.11** | **27.25** | **63.58** |
| **ViT-B/32** | | | | | |
| Zero-Shot CLIP | 59.88 | 51.98 | 39.22 | 27.44 | 63.79 |
| CLIP + CoOp ($M=16$) | **64.92** | 55.90 | 38.79 | 28.77 | 63.45 |
| CLIP + CoOp ($M=4$) | 64.88 | **56.21** | **40.17** | **29.64** | **64.60** |
| **ViT-B/16** | | | | | |
| Zero-Shot CLIP | 64.71 | 58.71 | 44.77 | 43.37 | 72.49 |
| CLIP + CoOp ($M=16$) | **70.13** | 62.23 | 44.82 | 44.30 | 72.98 |
| CLIP + CoOp ($M=4$) | 70.11 | **62.66** | **46.27** | **45.46** | **74.33** |

Table 7: Comparison with prompt engineering and prompt ensembling on ImageNet using different vision backbones.

| Method | ResNet-50 | ResNet-101 | ViT-B/32 | ViT-B/16 |
|---|---|---|---|---|
| Prompt engineering | 55.41 | 58.72 | 59.88 | 64.71 |
| Prompt ensembling | 57.81 | 60.49 | 62.01 | 67.31 |
| CoOp | **60.46** | **64.39** | **64.92** | **70.13** |

## B.4 VISION BACKBONES

Figure 7 provides the detailed per-dataset results for various vision backbones. The more advanced the backbone, the better the performance.

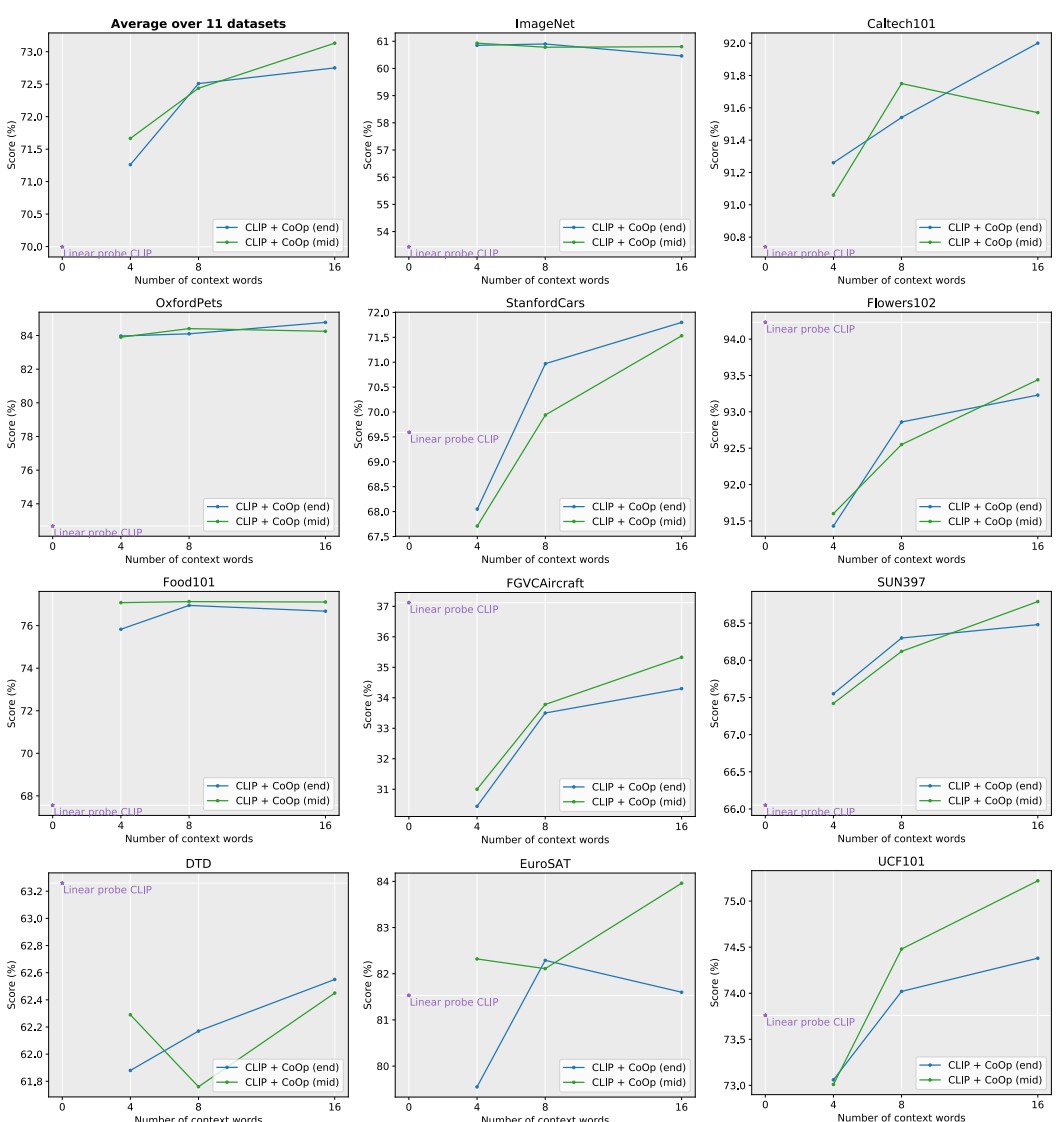

Figure 6: Dataset-specific results of using different context lengths for CoOp.

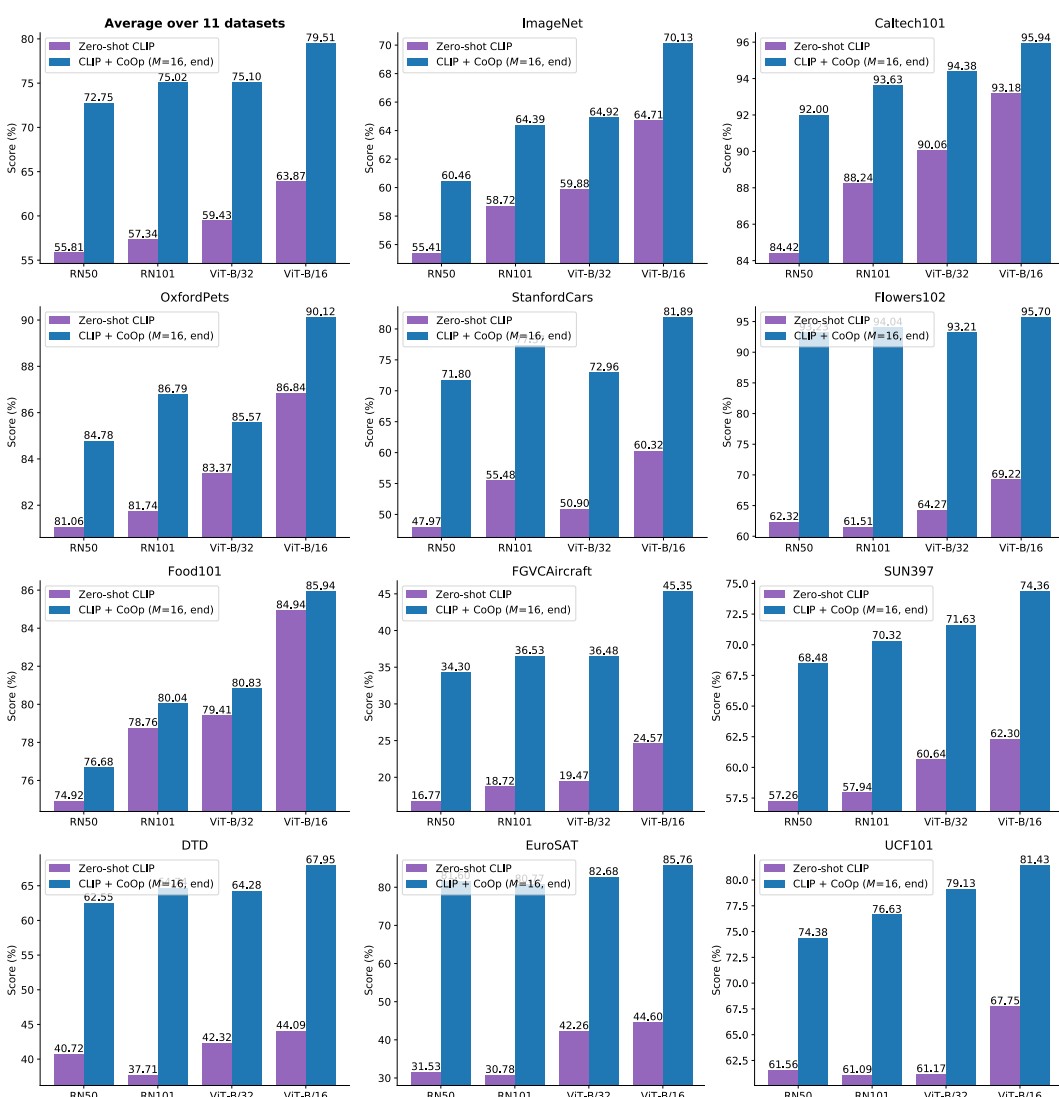

Figure 7: Results on the 11 datasets using a variety of vision backbones.

