# OpenReview forum: "Learning to Prompt for Vision-Language Models"
_ICLR.cc/2022/Conference — ICLR 2022 Submitted_

### Official Review · Reviewer_KMdr · 2021-10-21

**Correctness:** 3
**Technical Novelty And Significance:** 2
**Empirical Novelty And Significance:** 2
**Recommendation:** 1
**Confidence:** 5

**Main Review:**

+ Extensive experiments
+ Motivation for the automatic prompt is useful

o Investment in Fig 1 is super useful. It is helpful to see how fragile the prompt is affected by word twisting. However, the paper claimed prompt engineering is time-consuming, which I do not disagree with. Yes, human annotators have no way to know whether adding a "flower" or changing the sentence order will improve or harm the performance. But the key motivation of the prompt is to reduce the human working load and adding some general description is not super complicated. The task should be down to exploring more robust models and algorithms that can withstand these minor changes in my opinion.

- The biggest concern is about the lack of theoretical contribution. The main content of the methodology is broken down into two sections. The first section is just a review of CLIP which is already well-known. The key content of context optimisation in 2.2 is just half a page. The description is rough.

- It is not super clear why adding extra dimensions (claimed as the same number in that of word embedding) to the class token can help the prompt.

- On page 4 above Eq.3, the concept "unified context" and "class-specific context" are not well explained.

- paragraph under Eq.3, "Training is performed ..." is redundant and not informative.

- Figure 2 is not helpful in understanding the framework and looks very low-quality.

**Summary Of The Paper:**

This paper proposes a context optimisation (CoOp) approach that can automate prompt engineering and allow more efficient and task-specific transfer for pretrained vision-language models.

**Summary Of The Review:**

Despite extensive evaluation, the paper presents in very low quality and lack of description of key methods and the theory behind it. The overall quality is clearly below the threshold of the ICLR standard.

---

> ### Author Response · Authors · 2021-11-18
> **Response**
>
> Thank you for your review. We think designing efficient adaptation methods for large pre-trained vision-language models is an important step to democratize foundation models [a], which is an emerging topic in computer vision. We’d like to emphasize that *our paper is the first to tackle this problem in computer vision using prompt learning*, and hope our paper can be assessed fairly based on whether the insights and findings presented in the paper are novel to the field of computer vision.
>
> [a] Bommasani, R., Hudson, D. A., Adeli, E., Altman, R., Arora, S., von Arx, S., ... & Liang, P. (2021). On the opportunities and risks of foundation models. arXiv preprint arXiv:2108.07258.
>
> We address your concerns below.
>
> > However, the paper claimed prompt engineering is time-consuming ... The task should be down to exploring more robust models and algorithms that can withstand these minor changes in my opinion.
>
> Automatic prompt learning has been an established research problem in NLP (see [b] for a comprehensive survey in this topic) where the motivation has been widely recognized: prompt engineering needs labeled data and a significant amount of efforts for words tuning but does not guarantee that the resulting prompts are familiar to what the model has been learned with; so the idea is why not design an efficient algorithm to leverage the labeled data for automatic prompt learning.
>
> We think the adaptation of large pre-trained vision-language models is a promising research direction, which is of great importance to facilitating deployments of these models in downstream applications. Being *the first to study efficient prompt learning in computer vision*, our paper provides timely insights, and importantly, demonstrates strong empirical performance for a simple approach, which we think is worth sharing with the community.
>
> [b] Liu, P., Yuan, W., Fu, J., Jiang, Z., Hayashi, H., & Neubig, G. (2021). Pre-train, prompt, and predict: A systematic survey of prompting methods in natural language processing. arXiv preprint arXiv:2107.13586.
>
> > The biggest concern is about the lack of theoretical contribution.
>
> Our prompt learning-based approach has a very simple and intuitive design, i.e., to turn context words in a prompt into a set of learnable vectors. From the learning perspective, our approach is based on supervised learning, or more formally empirical risk minimization, which is backed by the statistical learning theory [c] well known to the machine learning community. Therefore, having a theory seems trivial for prompt learning methods. To our knowledge, none of the existing prompt learning papers in NLP has a theory (please refer to the relevant references in Section 4).
>
> [c] Vapnik, V. (1999). The nature of statistical learning theory. Springer science & business media.
>
> > The main content of the methodology is broken down into two sections. The first section is just a review of CLIP which is already well-known. The key content of context optimisation in 2.2 is just half a page. The description is rough.
>
> The purpose of putting a review of CLIP is to introduce the model architecture—on top of which our model is built—and the prompting-based zero-shot inference. We think the organization is clear and useful for readers who might not know CLIP’s technical details.
>
> Our approach is quite simple and easy to understand, particularly for readers who are already familiar with CLIP. Therefore, we chose to leave more space for the experiments.
>
> > It is not super clear why adding extra dimensions (claimed as the same number in that of word embedding) to the class token can help the prompt.
>
> Adding more tokens can increase the capacity of context, which is analogous to adding more parameters/layers in a neural network.
>
> > On page 4 above Eq.3, the concept "unified context" and "class-specific context" are not well explained.
>
> The explanation for “unified context” appears right before it, “Note that the context here is shared among all classes, which is called unified context ...”
>
> Class-specific context is explicitly explained in the last paragraph of Section 2, “Another option is to design class-specific context …”
>
> > paragraph under Eq.3, "Training is performed ..." is redundant and not informative.
>
> The paragraph is necessary because its purpose is to discuss how the model is trained as well as to explain the intuition why the design helps learn “meaningful” context, which we think would give readers a clearer picture of our approach.
>
> > Figure 2 is not helpful in understanding the framework and looks very low-quality.
>
> The figure is inspired by OpenAI’s sketch of the CLIP model and designed to be as simple as possible. Could you please give more specific comments about which parts you think are unclear or can be improved? We are happy to revise the figure.

---

> > ### Author Response · Authors · 2021-11-18
> > **Response**
> >
> > > ​​Despite extensive evaluation, the paper presents in very low quality and lack of description of key methods and the theory behind it. The overall quality is clearly below the threshold of the ICLR standard.
> >
> > We hope our answer to your main concern about the theory is resolved. The paper was carefully organized and polished, which is also endorsed by other reviewers who found our paper “generally well-written and easy to follow.” Any more advice that you think could improve the paper is welcome. We think the “strong reject” rating isn’t a fair assessment and hope you can re-evaluate our paper based on whether it has positive impacts on computer vision.

---

### Official Review · Reviewer_AYDN · 2021-10-27

**Correctness:** 4
**Technical Novelty And Significance:** 2
**Empirical Novelty And Significance:** 2
**Recommendation:** 5
**Confidence:** 3

**Main Review:**

The paper's motivation is clear, experiments thorough, and results
convincing. In addition to the standard few-shot evaluations, I
appreciated the authors considering the distribution shift scenario:
in that case, they show that their model can more effectively leverage
labelled data from a different dataset versus the linear probes (even
though zero-shot CLIP remains somewhat competitive in that
regime). Additional experiments about the optimal context length,
placement of fine-tune-able token embeddings, and vision backbones
where appreciated.

While the CoOp method appears to work well, but I had some concerns
about novelty. In particular, this method is more-or-less identical to
Li and Liang (2021) --- that paper came out on arXiv Jan 1 of this
year, and was published at ACL earlier this year. In addition, this
idea has been "rediscovered" in the NLP context several times (which
are also cited, but perhaps cannot be considered contemporaneous,
given their arXiv dates...). While the authors cite the arXiv version,
I think that, given the timing, the authors should perhaps not pitch
their method as a "novel" approach (as is done in the abstract);
rather, this seems to be applying prefix-tuning to CLIP.

I had a few technical concerns:

1. there are a lack of details for the linear probe --- how was the
   regularization parameter chosen?

2. The baseline for comparison to CoOp was the linear probe, which
   makes sense. However, I would have also liked to have seen a
   comparison where all the parameters of CLIP are fine-tuned --- how
   well does that work for few shot learning?

I also had a few presentation concerns:

1. Figure 1 compares a supervised CoOp method to a zero-shot CLIP
   baseline. While the CoOp method is "few shot", in this figure,
   there are 16 examples per class provided, which, for some datasets
   may amount to hundreds or thousands of labelled examples. I would
   have appreciated Figure 1 comparing to the linear probe.

2. Figure 5b has a similar problem with being potentially misleading:
   I would recommend including not only zero-shot CLIP, but also
   linear probe CLIP.

**Summary Of The Paper:**

The authors demonstrate a more efficient form of few-shot learning
using CLIP compared to linear probing for image classification: CoOp.
Instead of fine-tuning a small linear classifier on the output of
CLIP, they propose fine-tuning a number of additional embeddings at
the input layer; this modification, in theory, allows CLIP to leverage
more computation when adapting to tasks, while still only optimizing a
small number of parameters. Experiments across several corpora
demonstrate the efficacy of the approach, which generally yields a
few accuracy points of gain versus a linear probe trained on the same
amount of data.

**Summary Of The Review:**

Overall: the work is generally clear with thorough and convincing
experiments. While there were a few presentation/technical concerns,
the main drawback of this work is novelty: the proposed idea is
identical to Li and Liang (2021) [arxiv in january], Zhong et
al. (2021) [arxiv in April], and perhaps Lester et al (2021) [arxiv in
Sep, this one is more recent and I haven't read it yet]. While these
papers consider only the NLP case and not the vision+language case,
it's still difficult to call this method "novel," as the authors do in
the intro.

---

> ### Author Response · Authors · 2021-11-18
> **Response**
>
> Thank you for your review. Also thank you for endorsing our paper’s motivation and the empirical results. We are very encouraged by the comments.
>
> As the concern over the novelty was also raised by other reviewers, we provide a comprehensive discussion in the common area.
>
> We address your remaining concerns below.
>
> > There are a lack of details for the linear probe --- how was the regularization parameter chosen?
>
> As mentioned in the paper, we followed the same steps as in Radford et al. (2021). Please see Appendix A.3 in their original paper at https://arxiv.org/pdf/2103.00020.pdf.
>
> > The baseline for comparison to CoOp was the linear probe, which makes sense. However, I would have also liked to have seen a comparison where all the parameters of CLIP are fine-tuned --- how well does that work for few shot learning?
>
> Good question. We did tried this experiment early on where we fine-tuned all parameters in the image and text encoders. We found that this fine-tuning method did not work at all for all CLIP’s (open-source) vision backbones, i.e., RN50, RN101, ViT-B/32 and ViT-B/16. Specifically, the loss initially declined for a few iterations but then quickly went up and stayed at a “dead” mode with zero training accuracy.
>
> > Figure 1 compares a supervised CoOp method to a zero-shot CLIP baseline. While the CoOp method is "few shot", in this figure, there are 16 examples per class provided, which, for some datasets may amount to hundreds or thousands of labelled examples. I would have appreciated Figure 1 comparing to the linear probe.
>
> Please note that Fig.1’s purpose is not to show a comprehensive comparison of all methods (which is detailed in the experiments section, specifically in Sec.3.1 and 3.2) but to highlight the problems of hand-crafted prompts, explaining why we should learn the prompts rather than manually design them.
>
> > Figure 5b has a similar problem with being potentially misleading: I would recommend including not only zero-shot CLIP, but also linear probe CLIP.
>
> Please note that Fig.5b is part of the ablation studies, which mainly aims to verify whether the learning-based prompts work with different architectures. Sec.3.1 (few-shot learning) and 3.2 (domain generalization) have provided comprehensive comparisons with linear probe CLIP where the results are sufficient to justify the learning-based prompts. So repeating the experiments for linear probe CLIP across all 11 datasets and vision backbones would not give extra new insights but will certainly consume a lot of compute resources and time.

---

### Official Review · Reviewer_1WJt · 2021-11-03

**Correctness:** 3
**Technical Novelty And Significance:** 2
**Empirical Novelty And Significance:** 3
**Recommendation:** 5
**Confidence:** 4

**Main Review:**

Strengths:
1) The paper is generally well-written and easy to follow. The author conducts extensive experiments and ablation studies.
2) CoOp outperforms CLIP on all 11 diverse datasets, especially showing large improvements in specific domains like texture and satellite images.
3) CoOp is data-efficient and can boost the classification performance with only a few shots of training data.
4) CoOp demonstrates better robustness to distribution shift than zero-shot CLIP and linear probe CLIP.

Weakness:
1) Though effective, the technical novelty of this paper is quite limited. Since the soft prompt tuning approaches (e.g., Prompt Tuning [1] and P-Tuning [2]) have already been proposed in NLP domain. And CoOp adopts a very similar technique.
[1] Lester, Brian, Rami Al-Rfou, and Noah Constant. “The power of scale for parameter-efficient prompt tuning.” EMNLP 2021.
[2] Liu, Xiao, et al. “GPT Understands, Too.” *arXiv preprint arXiv:2103.10385* (2021).

2) According to Table 4 in the paper, the nearest neighbor words of the learned context vectors rarely have practical semantic meaning. This casts doubt on using the word “context”. From this perspective, these soft prompts are just more parameters to improve the model capacity.  Hence, can we say that CoOp is just a better version of fine-tuning? In other words, there might be a similar fine-tuning way to utilize additional parameters to get better performance.

**Summary Of The Paper:**

The paper proposes context optimization (CoOp) which learns task-aware continuous prompts to improve CLIP in terms of few-shot image classification. By fixing the pretrained backbone, CoOp performs end-to-end learning to update the learnable context vectors for target domain datasets. The simple yet effective approach substantially beats hand-crafted prompts with a large margin. Meanwhile, CoOp also exhibits better robustness to distribution shift than CLIP.

**Summary Of The Review:**

The paper achieves better few-shot image classification performance improvements but lacks enough technical novelty and explanations for learned prompts

---

> ### Author Response · Authors · 2021-11-18
> **Response**
>
> Thank you for your review. Also thank you for endorsing the empirical results. We are very encouraged by the comments.
>
> We address your concerns below.
>
> > Though effective, the technical novelty of this paper is quite limited. Since the soft prompt tuning approaches (e.g., Prompt Tuning [1] and P-Tuning [2]) have already been proposed in NLP domain. And CoOp adopts a very similar technique. [1] Lester, Brian, Rami Al-Rfou, and Noah Constant. “The power of scale for parameter-efficient prompt tuning.” EMNLP 2021. [2] Liu, Xiao, et al. “GPT Understands, Too.” arXiv preprint arXiv:2103.10385 (2021).
>
> As this concern was also raised by other reviewers, we provide a comprehensive discussion in the common area.
>
> > According to Table 4 in the paper, the nearest neighbor words of the learned context vectors rarely have practical semantic meaning. This casts doubt on using the word “context”. From this perspective, these soft prompts are just more parameters to improve the model capacity. Hence, can we say that CoOp is just a better version of fine-tuning? In other words, there might be a similar fine-tuning way to utilize additional parameters to get better performance.
>
> As discussed in the paper, interpreting the learned tokens is difficult and using nearest words from the pre-trained embeddings might be inaccurate (so we do not recommend future work to use this interpretation method). Since the tokens are updated using knowledge (i.e., gradients) distilled from the pre-trained text encoder, it is reasonable to think that the learned tokens still reside within the pre-trained word space but contain more abstract meanings that are useful for downstream recognition.
>
> We would also like to point out that the interpretation issue is not unique to CoOp but common to broader continuous prompt learning methods developed in NLP (which is discussed in the related work section). To our knowledge, none of the existing continuous prompt learning papers in NLP has solved the interpretation issue. We hope our openness to discuss the issue is encouraged rather than criticized as a weakness.
>
> CoOp is the first prompt learning-based adaptation method for CLIP-like foundation models [a] *in computer vision*, but certainly not the only workable solution. We think our simple design and the findings presented in the paper can be useful for the community to investigate more advanced, and effective, designs for adapting large pre-trained vision-language models—which we view as an important research direction to democratize foundation models [a].
>
> [a] Bommasani, R., Hudson, D. A., Adeli, E., Altman, R., Arora, S., von Arx, S., ... & Liang, P. (2021). On the opportunities and risks of foundation models. arXiv preprint arXiv:2108.07258.

---

### Official Review · Reviewer_pGwp · 2021-11-03

**Correctness:** 3
**Technical Novelty And Significance:** 3
**Empirical Novelty And Significance:** 3
**Recommendation:** 6
**Confidence:** 5

**Main Review:**

Strengths:
1. This paper provides a novel approach named CoOp for prompt engineering based on  CLIP.
2. CLIP-based experiments are sufficient to prove the advantages of CoOp.

Weaknesses:
1. Only the results based on CLIP are compared, and there are no more experiments of other visual-language models.
2. It is an improvement of CLIP, and the idea is similar to many existing works such as [1].
[1] Prefix-Tuning: Optimizing Continuous Prompts for Generation


**Summary Of The Paper:**

This paper proposes a novel approach named context optimization (CoOp) for prompt engineering of vision-language pre-training models.   The main idea is to model context in prompts using continuous representations and perform end-to-end learning from data while keeping the pre-trained parameters fixed. Experiments on 11 datasets show that CoOp effectively turns pre-trained vision-language models into data-efficient visual learners, requiring as few as one or two shots to beat hand-crafted prompts with a decent margin and able to gain significant improvements when using more shots.

**Summary Of The Review:**

This is a somewhat novel but solid work. The comparative experiment based on clip is very sufficient, but it also lacks other important experiments, such as the effect of CoOp on other vision-language models, just as the title of the paper is learning to prompt for vision language models

---

> ### Author Response · Authors · 2021-11-18
> **Response**
>
> Thank you for your review. We are very encouraged by your positive comment on the solidity of our paper.
>
> We address your concerns below.
>
> > Only the results based on CLIP are compared, and there are no more experiments of other visual-language models.
>
> First, we would like to clarify that we refer to vision-language models specifically as recent *CLIP-like image recognition models*, which generate classification weights directly from natural language. This concept was used in Jia et al. (2021).
>
> To our knowledge, only CLIP (Radford et al., 2021) is open-source and has been widely used by the community. We expect our approach to also work for other similar models like ALIGN (Jia et al., 2021) and the more recent DeCLIP [a] and CLOOB [b].
>
> [a] Li, Y., Liang, F., Zhao, L., Cui, Y., Ouyang, W., Shao, J., ... & Yan, J. (2021). Supervision Exists Everywhere: A Data Efficient Contrastive Language-Image Pre-training Paradigm. arXiv preprint arXiv:2110.05208.
>
> [b] Fürst, A., Rumetshofer, E., Tran, V., Ramsauer, H., Tang, F., Lehner, J., ... & Hochreiter, S. (2021). CLOOB: Modern Hopfield Networks with InfoLOOB Outperform CLIP. arXiv preprint arXiv:2110.11316.
>
> > It is an improvement of CLIP, and the idea is similar to many existing works such as [1]. [1] Prefix-Tuning: Optimizing Continuous Prompts for Generation.
>
> As this concern was also raised by other reviewers, we provide a comprehensive discussion in the common area.

---

### Author Response · Authors · 2021-11-18
**Response to the concern about novelty**

Since most reviewers share the same concern about the novelty that the idea of prompt learning has been proposed in NLP, we provide our response in this common area and hope our answer can convince the reviewers to support our paper.

First of all, our research deals with *computer vision* problems that are significantly different from those in NLP so we hope our paper can be assessed based on whether the insights and findings are useful to the vision community. To avoid misunderstanding, we’ve changed the word “novel” to “simple” in the abstract when describing the proposed approach.

Successfully applying ideas developed in a different domain like NLP to computer vision is *non-trivial* and could inspire new ideas that might not be possible by only focusing on technical innovation in a single field. A recent example is Vision Transformers [a], which applies the same Transformers [b] architecture developed in NLP to image recognition and has now been extended to many other computer vision problems.

We are the first to successfully apply the concept of prompt learning to computer vision, specifically to adapting large pre-trained vision-language models, which is an important topic essential for democratizing foundation models [c] (i.e., to make these models more accessible to wider communities that do not have sufficient training data and compute resources for pre-training). Technically, we propose two variants of prompt learning: unified context optimization and class-specific context optimization, which are specifically designed for solving image recognition problems. We believe the insights and findings presented in the paper are novel to computer vision and worth sharing with the community.

[a] Dosovitskiy, A., Beyer, L., Kolesnikov, A., Weissenborn, D., Zhai, X., Unterthiner, T., ... & Houlsby, N. (2020). An image is worth 16x16 words: Transformers for image recognition at scale. In ICLR, 2021. https://openreview.net/forum?id=YicbFdNTTy

[b] Vaswani, A., Shazeer, N., Parmar, N., Uszkoreit, J., Jones, L., Gomez, A. N., ... & Polosukhin, I. (2017). Attention is all you need. In Advances in neural information processing systems (pp. 5998-6008).

[c] Bommasani, R., Hudson, D. A., Adeli, E., Altman, R., Arora, S., von Arx, S., ... & Liang, P. (2021). On the opportunities and risks of foundation models. arXiv preprint arXiv:2108.07258.

---

> ### Comment · Reviewer_AYDN · 2021-11-24
> **Thanks for the info!**
>
> Hi there --- just confirming that I saw this and am considering your response in my continued capacity as a reviewer for this work; thank you for detailing your thoughts.

---

### Decision · Program_Chairs · 2022-01-20

**Decision:**

Reject

**Comment:**

Given the increasing scale of large models (e.g. CLIP), there's an argument that we need better automated techniques for properly utilizing (prompting) these models. Given the success of prompt learning within pure NLP models, the authors apply the same approach to the V+L domain and show that it also is applicable here.  Generally, reviewers felt that the results were clear and thorough, yet technically limited.  The approach is not novel and the result not surprising.  There is a documentary benefit to having this work out in the community for others to reference and extend.